



# Intense wind-driven coastal upwelling in the Balearic Islands in response to storm Blas (November 2021)

Baptiste Mourre[1], Emma Reyes[1], Pablo Lorente[2], Alex Santana[1], Jaime Hernández-Lasheras[1], Ismael Hernández-Carrasco[1], Maximo García-Jove Navarro[1], Nikolaos D. Zarokanellos[1]

[1]SOCIB, Balearic Islands Coastal Observing and Forecasting System, Palma, 07122, Spain
[2]Puertos del Estado, Madrid, 28042, Spain

*Correspondence to: Baptiste Mourre (bmourre@socib.es)*

**Abstract.**

This article analyzes the Balearic Islands wind-driven coastal upwelling in response to the intense and long-lasting storm Blas which affected the Western Mediterranean Sea in November 2021. The storm was associated with a pronounced pressure low,

generating heavy rains and intense winds and showing some characteristics of a tropical cyclone. The Balearic Islands area was particularly affected since the core of the storm was moving over a one-week-long period from the south-west of this area to just above Menorca and Mallorca Islands. High-resolution regional forecast models indicated that the intense northeasterly winds blowing over the region during the first days of the storm led to the development of intense upwellings along the northwestern coasts of Mallorca and Ibiza Islands together with a reversal of the surface current. While the clouds associated

with the storm prevented the radiometers onboard satellites to precisely observe the evolution of the sea surface signature of the upwelling, signals of enhanced chlorophyll concentration were still detected in the upwelling region. The high-resolution Western Mediterranean Operational model, which downscales the Copernicus Marine Service Mediterranean predictions, is used to describe the characteristics of this intense coastal upwelling event, as well as to analyze its singularity over the past 9-year time series through the comparison of different coastal upwelling indices. The upwelling event is found to have a duration

of three days with a spatial off-shore extension of around 20km. It was characterized by intense cold coastal sea surface anomalies of around 6ºC. While it was the most intense event over the past 9 years in terms of local cross-shore sea surface temperature gradients, it is ranked second in terms of the intensity of cross-shelf transports, following the Gloria episode in January 2020.


## Short summary

We characterize the surface and vertical signature of an intense storm-induced coastal upwelling along the northwestern coast of the Balearic Islands in 2021, using a high-resolution numerical model. The upwelling, with a duration around three days





and a spatial off-shore extension of 20km, led to cross-shore surface temperature differences of up to 6ºC. It was the most

intense event of the past 9 years in terms of the impact on temperature, the second one considering cross-shore transports.

## 1- Introduction

Storm Blas[1] was an intense Mediterranean cyclone which affected the Western Mediterranean Sea from 6 to 18 November 2021. It was first identified on 6 November between the Balearic Islands and Sardinia before moving westwards towards the

Balearic archipelago. After the core of the storm looped over Mallorca Island on 11 November, exhibiting a well-defined circular deep (~10hPa) low-pressure center, it then moved eastwards while developing a spiral structure resembling that of tropical cyclones. The storm then moved over Sardinia and Corsica Islands before weakening and dissipating in the Tyrrhenian Sea on 18 November. Panels a and b in Figure 1 show the distribution of clouds over the affected area on 7 and 13 November as observed by Sentinel-3 Ocean and Land Color Instrument (OLCI) satellite true color images.

This situation created extreme conditions of intense winds and high waves as well as heavy rainfall in the Balearic archipelago. In particular, intense northeasterly winds were blowing over the area during the first phase of the storm from 5 to 7 November (Figure 1 panels c,d), creating favorable conditions for coastal upwellings along the northwestern coasts of Mallorca and Ibiza Islands. The peak of this wind event occurred on 6 November around 10:00 UTC, with hourly mean values over 17.5 m/s off Mallorca Island as represented by the HARMONIE-AROME model (Bengtsson et al., 2017) from the Spanish Meteorological

Agency (AEMET).

Coastal upwellings are ocean processes which generate upward vertical currents at the coast in response to an offshore transport of surface waters produced under the action of an intense or sustained alongshore wind stress. In the northern hemisphere, the wind has to blow with the coast on its left-hand-side to be favorable to upwelling. By bringing cold and nutrient-rich deep waters close to the surface, upwellings have significant effects on the physical and biogeochemical characteristics of coastal

waters, generally enhancing the local primary productivity and then playing an important role on marine ecosystems (Pauly and Christensen, 1995). In the Western Mediterranean Sea, wind-induced upwellings are known to occur along the French coast (Millot et al, 1979, Ray et al. 2000, Bakun and Agostini, 2001), western and eastern coasts of Sardinia (Olita et al, 2013, Salusti et al., 1998) and northwestern Alboran Sea (Sarhan et al., 2000, Macias et al., 2008). Eddy-induced coastal upwellings have also been evidenced in the Ligurian Sea (Casella et al., 2011) and along the Algerian coast (Millot et al., 1985). To our

knowledge, no upwelling has yet been reported in the Balearic Islands. While the frequent north-northwesterly mistral wind events do not present the appropriate orientation to generate upwellings along the northwestern coast of the Balearic Islands, favorable conditions with intense northeasterly winds are occasionally met during specific storms such as Gloria in January 2020 (Amores et al., 2020, Sotillo et al., 2021, Álvarez-Fanjul et al. 2022) or Blas in November 2021. Notice that these winds

---

[1] https://www.aemet.es/en/conocermas/borrascas/2021-2022/estudios_e_impactos/blas





blow in the opposite direction of the established Balearic Current, which flows northeastwards along the northwestern coast of the Balearic Islands (Lopez-García et al., 1994).

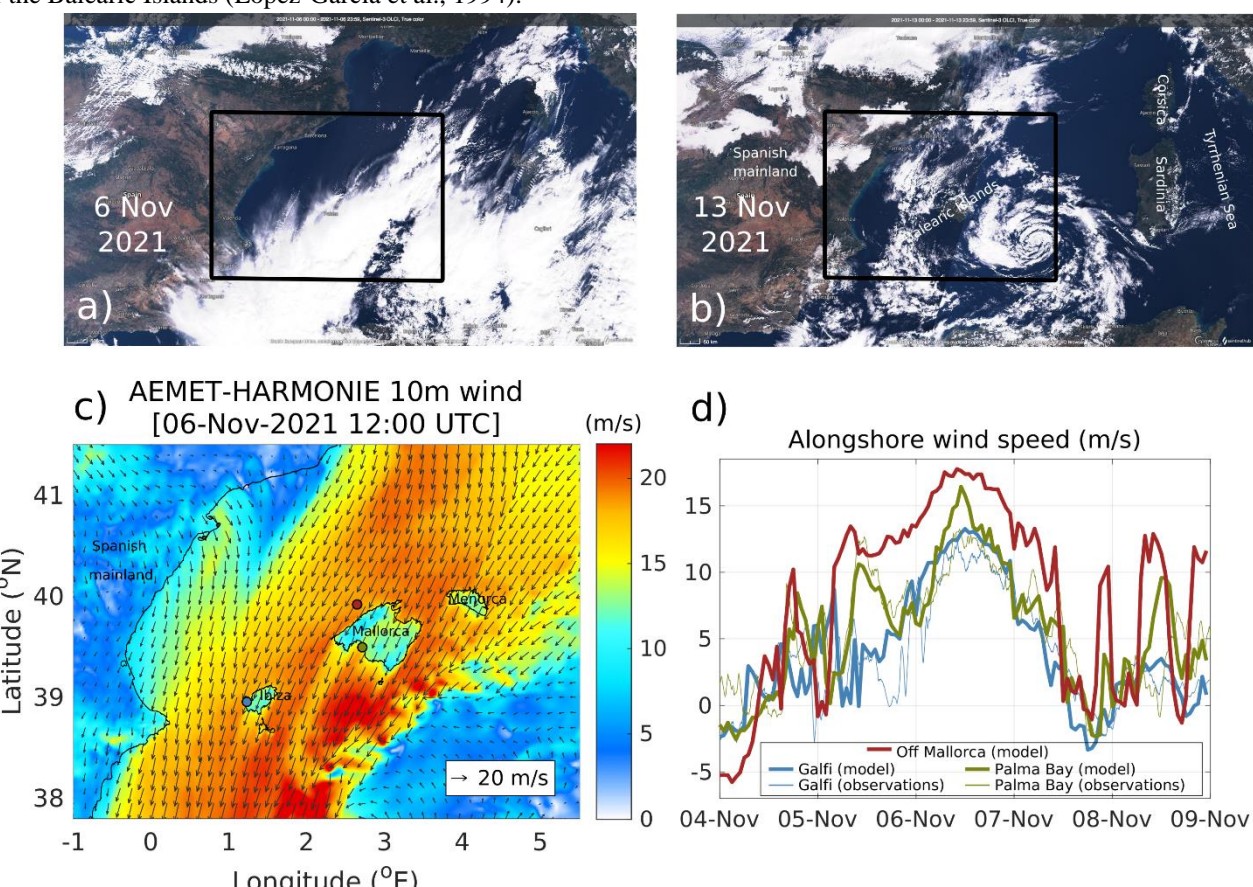

**Figure 1: Upper panels: True color image from Sentinel-3 OLCI captured over the Western Mediterranean Sea on a) 6 November 2021 and b) 13 November 2021. Lower panels: c) 10m wind map on 6 November 2021 from the AEMET HARMONIE-AROME prediction model over the area delimited by the black rectangles in panels a) and b); and d) time series of alongshore 10m wind speed from model and observations at the three locations (1-Automatic Weather Station of Puig des Galfí in Ibiza, 2- Oceanographic**
**buoy of Palma Bay in Mallorca, and 3- virtual station off the northwestern coast of Mallorca) represented in panel c). "Alongshore" is here defined considering the direction of the northwestern coast of Mallorca Island making an angle of 123º with respect to the North, with positive values southwestwards.**

This study originates from the representation of intense sea surface temperature (SST) gradients along the northwestern coast of the Balearic Islands in high-resolution regional prediction models during the first phase of storm Blas, together with
observations of surface current reversal from High-Frequency (HF) radar measurements along the northwestern coast of Ibiza Island. The objective of this study is to characterize this unusually intense upwelling in terms of both its surface signature and its vertical structure, as well as to evaluate its singularity in the perspective of the time series of different upwelling indices computed over the past decade. The main source of information is a high-resolution numerical prediction model given the very limited number of in-situ and satellite observations available during this event.






## 2- Data and methods

### 2.1 Numerical models

We analyze the outputs of the Western Mediterranean Operational prediction system (WMOP[2], Juza et al., 2016; Mourre et al., 2018) developed at the Balearic Islands Coastal Observing and Forecasting System (SOCIB, Tintoré et al., 2013). WMOP

is a 2-km regional configuration of the ROMS modelling system (Shchepetkin and McWilliams, 2005) implemented over the Western Mediterranean Sea. It uses high-resolution (1h, 2.5km) atmospheric forcing from the HARMONIE-AROME model (Bengtsson et al., 2017) provided by AEMET. Notice that before February 2019, the HIRLAM model was used as atmospheric forcing with a resolution of 5km and 1h (3h before March 2017). WMOP downscales the conditions of the Copernicus Marine Service Mediterranean analysis and forecast model (CMEMS-MED, Clementi et al., 2021), which are used as open boundary

conditions. It also includes assimilation of observations from satellite SST, along-track sea level anomaly, Argo temperature and salinity profiles as well as surface currents in the Ibiza channel applying a local multimodel Ensemble Optimal Interpolation approach as described in Hernández-Lasheras and Mourre (2018) and Hernández-Lasheras et al. (2022), with a 3-day cycle.

We also consider the outputs of the CMEMS-MED model, which provides a spatial resolution of around 4km. This system

includes variational data assimilation of temperature and salinity vertical profiles and along track satellite Sea Level Anomaly observations through a 3DVAR scheme. The atmospheric forcing is provided by the predictions from the European Center for Medium-Range Weather Forecasts with a spatial resolution close to 10km and a temporal resolution of 1h.

### 2.2 Ocean color satellite observations

High resolution ocean color imagery was analyzed to detect the enhancement of the surface chlorophyll-a (Chla) concentration

during the upwelling. The Level-3 ocean color product distributed by the Copernicus Marine Service (Volpe et al. 2019) was used. It provides surface Chla concentration from the OLCI Instrument onboard the Sentinel-3 satellite with a 300m-resolution. Notice that clouds were present over the study area during the storm period, which limited the availability of exploitable high-resolution satellite observations. The Sentinel3-OLCI observations were the only available satellite data that were found to give relevant information for the detection of the upwelling signature during the study period.

### 2.3 Upwelling indices

Several upwelling indices (UIs) have been used in the literature with the objective to estimate the intensity of coastal upwellings. These indices are based on either the cross-shore Ekman transport related to the forcing winds (Bakun, 1973),

---

[2] https://www.socib.es/?seccion=modelling&facility=forecast

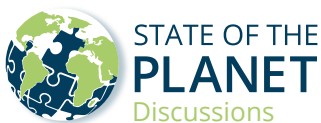

sometimes also corrected from the effect of onshore geostrophic flows (Marchesiello and Estrade, 2010; Rossi et al., 2013), the along-shore surface current velocities (Lorente et al., 2020), the estimates of total vertical transport and nitrate fluxes (Jacox et al., 2018) or the cross-shore SST differences (Demarcq and Faure, 2000). This study compares four UIs computed over the 9-year time series of the WMOP operational model outputs:

(i)  UI$_{SST}$ first quantifies the temperature differences between the coast ($SST_{coast}$) and a location 25km offshore ($SST_{offshore}$) off the northwestern coast of Mallorca Island (edges of the magenta cross-shore section illustrated in Figure 2a). As in Marchesiello and Estrade (2010), the observed differences are normalized by the vertical temperature differences between the surface and 200m depth ($T^{200m}$) at the offshore end of the section according to the following formula:

$$UI_{SST} = \frac{SST_{offshore} - SST_{coast}}{SST_{offshore} - T^{200m}_{offshore}} \qquad (1)$$

This normalized formulation allows to scale the horizontal temperature differences by the vertical ones. A value close to one indicates a fully developed upwelling with SST values at the coast equal to the temperatures at 200m depth. Since this formulation of UI$_{SST}$ generates very high values in situations when the upper ocean is vertically mixed during winter (denominator close to zero), the index is only computed here when the vertical temperature difference is larger than 1°C.

(ii)  The second index is based on alongshore surface current velocities, as first proposed in Lorente et al. (2020) as a proof-of-concept investigation. This index was specifically designed to be applied in any coastal area where surface velocities are available from HF radar instruments. It assumes that the alongshore wind stress is the primary driver of upwelling circulation and that the surface currents are highly responsive to local winds (e.g. Paduan and Ronsenfeld, 1996; Kohut et al., 2006). This index is defined here as the average alongshore surface velocity ($V^{surf}_{alongshore}$) in the 25km-wide coastal region off the northwestern coast of Mallorca Island, following Eq 2.

$$UI_{alongshore\ velocity} = \overline{V^{surf}_{alongshore}} \qquad (2)$$

The overbar denotes the average over the rectangular box containing the two magenta sections represented in Figure 2. The alongshore direction is the direction of the corresponding magenta section. It makes an angle of 123° with respect to the North. Positive values denote southwestwards flows. Further details about the methodology and application of this index can be found in Lorente et al. (2022, present issue of the Ocean State Report).

(iii)  The third index is the classical cross-shore Ekman transport index, computed from the alongshore wind stress. It is defined as follows:

$$UI_{Ekman\ transport} = \int_{x=0}^{x=L} \frac{\tau_{alongshore}}{\rho\ f} dx \qquad (3)$$





where $\tau_{alongshore}$ is the alongshore wind stress, $\rho$ is the water density and $f$ the Coriolis frequency. The integral is computed along the magenta alongshore section over a distance $L$ of 60km. $UI_{Ekman\ transport}$ represents the total transport across this section. Positive values indicate an offshore transport.

(iv)     Finally, the last index computes the total model transport in the upper 50m across the same section. This depth approximately corresponds to the depth of the Ekman layer as illustrated in Figure 3. It does not only include the Ekman transport, but also the cross-shore geostrophic transport and other contributions due to the cross-shore wind component and potential wind stress curl effects. It is calculated following Eq. (4):

$$UI_{total\ transport} = \int_{z=0}^{z=50m} \int_{x=0}^{x=L} V_{cross-shore}\ dx\ dz \qquad (4)$$

where $V_{cross\text{-}shore}$ is the model horizontal velocity in the cross-shore direction.



## 3- Surface expression of the upwelling



**Figure 2: Upper panels: SST and surface currents around Mallorca and Ibiza Islands on 7 November 2021 at 06:00 UTC as represented by a) WMOP and b) CMEMS-MED models. In panel a), the lines in magenta represent the alongshore and cross-shore sections used in Sections 3 and 4. Lower panels: chlorophyll-a concentration as observed by Sentinel3-OLCI on c) 4 November and d) 7 November (i.e. before and during storm Blas, respectively).**

Figure 2 illustrates the surface circulation patterns and associated SST in the WMOP and CMEMS-MED predictions models on 7 November 2022 at 06:00 UTC (the time with the most significant impact of the upwelling on the SST). The SST exhibits marked upwelling signatures in the WMOP model along the northwestern coast of Mallorca and Ibiza Islands, with the temperature of surface coastal waters around 6°C colder than that found 20km offshore. The surface coastal current flows southwestwards along the wind direction, which is reversed with respect to the direction of the Balearic Current under normal

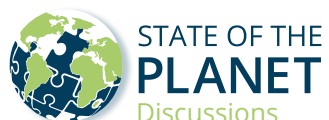

conditions (Lopez-García et al., 1994). The offshore width of the upwelled water region ranges from 10 to 20km, which slightly

extends off the 200m isobath. The upwelling is also represented in the CMEMS-MED model, but with a less pronounced signature. It is out of the scope of this short article to investigate these differences between models, but the lower spatial resolution of both the model grid and atmospheric forcing may have an important role in producing these discrepancies. The upwelled water only reaches the surface in the northern half of the northwestern coast of Mallorca Island in CMEMS-MED, yet with a larger offshore extension compared to WMOP. The event duration was around three days, from 5 November at

15:00 UTC when the first cross-shore coastal SST gradients were observed until 8 November at 12:00 UTC when they vanished.

As illustrated in Figure 1, the storm was associated with a dense cloud coverage, which only allowed very partial satellite remote sensing information on SST and Chla concentration. Despite this important limitation, the two most relevant ocean color images illustrated in Figure 2 provide indices of the enhancement of the Chla concentration along the northwestern

Mallorca coast. While a low concentration of Chla was detected on 4 November before the storm, a significant increase was observed on 7 November. The Chla concentration reached 0.1 mg/m3, a magnitude more than twice larger than that observed 3 days before. Moreover, the spatial distribution of Chla enhancement aligns well with the coastal pattern of cold waters in the WMOP model, suggesting a reasonable representation of the observed process in the model.

**3- Vertical structure**

Figure 3 illustrates the vertical structure of the temperature and velocity fields along the cross-shore section off the northwestern coast of Mallorca Island, as represented by the WMOP model. While velocities are illustrated on 6 November at 12:00 UTC, corresponding to the maximum wind intensity in this area (Figure 1), the temperature is displayed 18 hours later to illustrate the time with the maximum signature in SST. The temperature section (Figure 3a) shows a typical pattern of upwelling with an upward tilting of the isotherms towards the coast, which is associated with an upward transport of colder

deep waters along the topographic slope. The surface temperature decreases by 6ºC across the upwelling front, varying from 20.6ºC 20km offshore, where the mixed layer depth is around 50m, to 14.6ºC at the coast. The cross-shore velocity section (Figure 3b) shows an offshore displacement of surface waters in the upper 50m, consistent with the Ekman theory in the presence of alongshore winds. Cross-shore velocities (Figure 3c) reach 0.3 m/s. The alongshore velocity along this cross-shelf section is characterized by the intense southwestward coastal jet with surface velocities exceeding 1m/s. Upward vertical

velocities (Figure 3d) are enhanced along the topographic slope, with values between 40 and 60m/day in the upper 100m and in the first 12km from the coast. Overall, these cross-shelf sections illustrate a relatively thin (cross-shore extension around 20km) but intense upwelling with a significant coastal jet.




**Figure 3: Sections of WMOP model fields along the cross-shore magenta line represented in Figure 2: a) temperature on 7 November 2021 at 06:00 UTC, b) cross-shore velocity, c) alongshore velocity and d) vertical velocity on 6 November 2021 at 12:00 UTC.**

## 4- Interannual perspective

How singular was this intense wind-driven coastal upwelling event? To answer this question, we compute here several

upwelling indices (described in section 2.3) over the whole WMOP model time series covering the 9-year period between

August 2013 and June 2022. The first index, representing the normalized cross-shelf SST gradients, ranks the November 2021

event (i.e. Blas storm) as the most intense event of the whole time series. It is the only event with a normalized cross-shelf





SST gradient exceeding 0.5. The event is also the most intense according to the second index which measures the intensity of the alongshore surface velocity. However, the Ekman and total cross-shelf transport indices indicate a larger transport during the other extreme event associated with storm Gloria in January 2020 (Amores et al., 2020; Sotillo et al., 2021; Álvarez-Fanjul et al. 2022). This is especially true for the total transport which also accounts for the effects of the wind stress curl and cross-shore geostrophic transport. During both Gloria and Blas upwelling events, the offshore total transport was enhanced with respect to the Ekman estimate. A third very intense storm in January 2017 (also documented in García-León et al., 2018) led to a slightly larger cross-shore Ekman transport but a lower total transport compared to Blas. The effects on the SST and coastal jet are particularly marked during the Blas upwelling event due to its occurrence in early November when the surface ocean stratification was still significant. Despite the intense offshore transport during the upwelling event associated with storm Gloria, the effects on the SST were hardly noticeable due to the presence of mixed surface waters during this event in January 2020. Let´s remind here that the SST index is only computed when the vertical temperature difference is larger than 1°C, which significantly limits the use of the SST-based index during the winter season. However, this was not the case during the Gloria event, which showed vertical temperature differences just above 1°C.



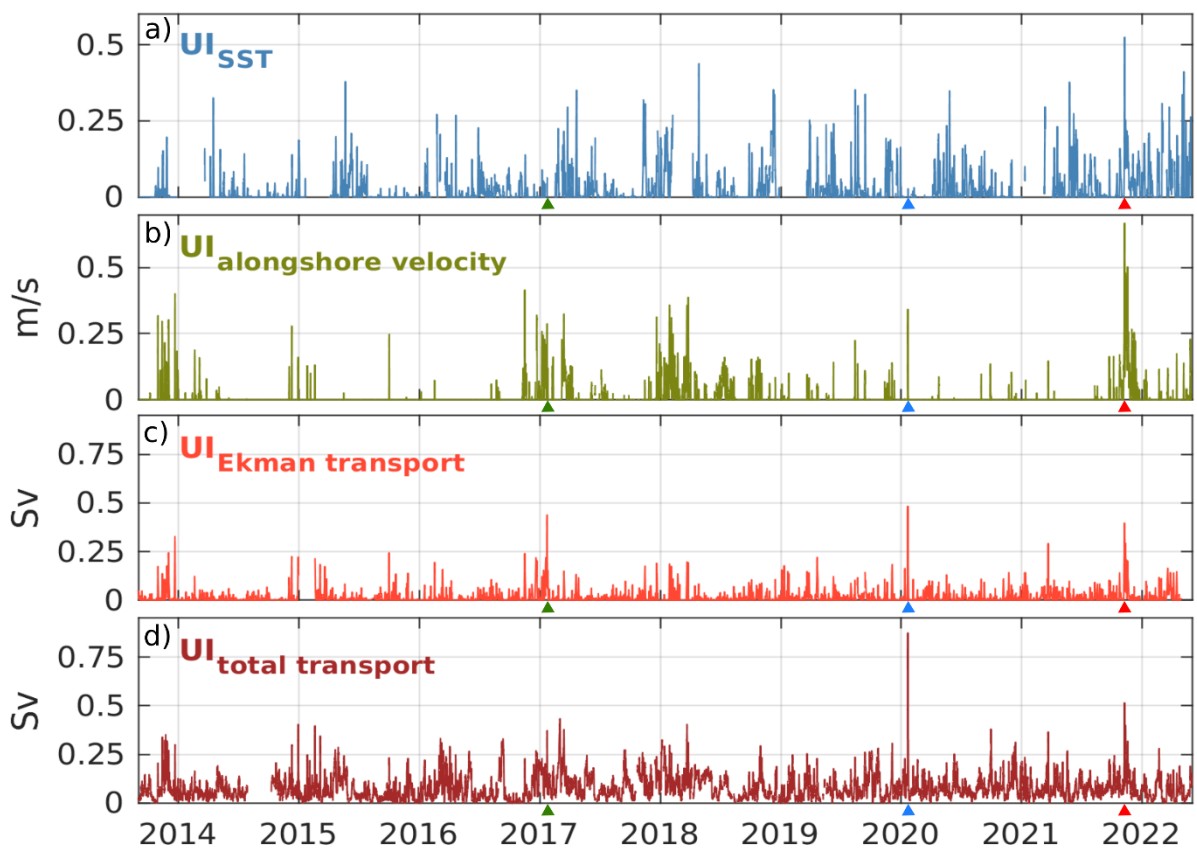

**Figure 4: Time series of daily upwelling indices defined in Section 2.3 between August 2013 and June 2022: a) cross-shore normalized SST gradient index, b) alongshore velocity index, c) cross-shore Ekman transport index, d) cross-shore total transport index. The red, blue and green triangles on the x-axis mark the time of Blas, Gloria and January 2017 storms, respectively.**

**5- Conclusions**

This study describes some of the characteristics of a short (5-8 November 2021) but intense wind-driven coastal upwelling event along the northwestern coast of the Balearic Islands as represented by a high-resolution forecast model during storm Blas in November 2021. The time series of several upwelling indices illustrate the episodicity of upwelling events in this area mainly related to their nature related to the occurrence of storms associated with intense northeasterly winds.

While the November 2021 Blas-related event was the most intense in terms of the effects on the SST and alongshore velocities over the 9-year-long analyzed time series, the induced cross-shelf surface transport was lower than that modelled during storm Gloria in January 2020, whose effects on the wind, waves, sea level and currents were already reported as a record-breaking





event in the literature (Álvarez-Fanjul et al. 2022). This comparison also demonstrates the complementarity between upwelling indices for the characterization of the intensity of these events.

Such storm-related upwellings are especially difficult to monitor given their short duration and the absence of in situ observations in this area. The dense cloud coverage associated with the storm also significantly limits the availability of high-resolution satellite observations to characterize the surface signature of the phenomenon. In this context, high-resolution forecast models provide very useful tools to overcome these limitations and describe the phenomenon. The only available ocean color image from satellite suggests an increase in primary productivity in the area of upwelled water as represented by

the model, providing relative confidence in the realism of the numerical results. High spatial resolution (close to kilometric) is needed in both the model grid and the atmospheric forcing to describe such coastal phenomena whose cross-shore extension does not exceed 20km.

Despite its short duration, this phenomenon was found to be sufficient to enhance the surface Chla concentration. A more comprehensive understanding of the impacts of these storm-related upwelling events on local ecosystems would need further

investigation requiring dedicated monitoring systems. Underwater gliders equipped with sensors of both physical and biogeochemical parameters can be suitable observation platforms for these studies given their capacity to operate under any weather conditions. In the future, the possible increase in the intensity and duration of the storms affecting the Western Mediterranean area (Gaertner et al., 2007, Romero and Emanuel, 2013, Gonzalez-Alemán et al. 2019) could lead to enhanced impacts of such storm-related upwelling events.

**Data availability**

The model and observation products used in this study from both the Copernicus Marine Service and other sources are listed in Table 1.

| Ref no. | Product name and type | Documentation |
|---|---|---|
| 1 | Images from the Copernicus Sentinel-3 Ocean and land mission. | Sentinel-3 Mission description: https://sentinels.copernicus.eu/web/sentinel/missions/sentinel-3 Data source: Data acquired by the Ocean and Land Colour Instrument (OLCI) and downlowaded from Sentinel Hub services (EO Browser), sentinel-hub.com |



| 2 | INSITU_MED_NRT_OBSERVATIONS_013_035 <br><br> [Mediterranean Sea- In-Situ Near Real Time Observations] | PUM: <br><br> https://catalogue.marine.copernicus.eu/documents/PUM/CMEMS-INS-PUM-013.pdf <br><br> QUID: <br><br> https://catalogue.marine.copernicus.eu/documents/QUID/CMEMS-INS-QUID-013-030-036.pdf |
|---|---|---|
| 3 | HARMONIE-AROME <br><br> [NWP -Numerical Weather Prediction- model] | System description: https://www.aemet.es/en/noticias/2017/07/modelo_harmonie-arome <br><br> Data source: AEMET. <br><br> Scientific references: Bengtsson et al., 2017. |
| 4 | SOCIB-WMOP <br><br> [Western Mediterranean OPerational model] | Product description: http://socib.es/?seccion=modelling&facility=forecast_system_description <br><br> Data access: <br><br> http://thredds.socib.es/thredds/catalog/operational_models/oceanographical/hydrodyn amics/wmop/catalog.html <br><br> References.: Juza et al., 2016; Mourre et al., 2018; Lasheras-Hernández and Mourre, 2018; Lasheras-Hernández et al., 2021 |
| 5 | MEDSEA_ANALYSIS_FORECAST_PHY_006_013 <br><br> [Mediterranean Sea Physics Analysis and Forecasting Product] | PUM: https://catalogue.marine.copernicus.eu/documents/PUM/CMEMS-MED-PUM-006-013.pdf <br><br> QUID: https://catalogue.marine.copernicus.eu/documents/QUID/CMEMS-MED-QUID-006-013.pdf |
| 6 | SST_MED_SST_L4_NRT_OBSERVATIONS_010_004 <br><br> [Mediterranean Sea High Resolution and Ultra High Resolution Sea Surface Temperature Analysis | PUM: https://catalogue.marine.copernicus.eu/documents/PUM/CMEMS-SST-PUM-010-004-006-012-013.pdf <br><br> QUID: https://catalogue.marine.copernicus.eu/documents/QUID/CMEMS-SST-QUID-010-004-006-012-013.pdf |





| 7 | SEALVEL_EUR_PHY_L3_NRT_OBSER VATIONS_008_059<br><br>[European seas along-track L3 sea level anomalies NRT tailored for data assimilation ] | PUM: http://marine.copernicus.eu/documents/PUM/CMEMS-SL-PUM-008-032-068.pdf<br><br>QUID: http://marine.copernicus.eu/documents/QUID/CMEMS-SL-QUID-008-032-068.pdf |
|---|---|---|
| 8 | INSITU_GLO_UV_NRT_OBSERVATIO NS_013_048<br><br>[Global Ocean- in-situ Near real time observations of ocean currents] | PUM:<br><br>https://catalogue.marine.copernicus.eu/documents/PUM/CMEMS-INS-PUM-013-048.pdf<br><br>QUID:<br><br>https://catalogue.marine.copernicus.eu/documents/QUID/CMEMS-INS-QUID-013-048.pdf |
| 9 | OCEANCOLOUR_MED_BGC_L3_NRT _009_141<br><br>[Mediterranean Sea Ocean Colour Plankton, Reflectance, Transparency & Optics L3 NRT daily observations] | PUM:<br><br>https://catalogue.marine.copernicus.eu/documents/PUM/CMEMS-OC-PUM.pdf<br><br>QUID:<br><br>https://catalogue.marine.copernicus.eu/documents/QUID/CMEMS-OC-QUID-009-141to144-151to154.pdf |

**Table 1: Products from the Copernicus Marine Service and other complementary datasets used in this study, including the Product User Manual and quality information. For complementary datasets, the link to the product description, data access and scientific references are provided.**

**Acknowlegements**

The authors thank AEMET, the Spanish Meteorological Agency, for providing the HARMONIE-AROME atmospheric fields to force the WMOP ocean model. We acknowledge SOCIB and Puertos del Estado for collecting oceanic and atmospheric observations around the Balearic Islands and distributing them through both their institutional websites and Copernicus database. This study would not have been possible without the data delivery from the Copernicus Marine Service.





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
