# Peer review of "Intense wind-driven coastal upwelling in the Balearic Islands in response to storm Blas (November 2021)"

_State of the Planet, 2022_

## Author Comment (AC2)

**Sentinel3-OLCI Quality Index (QI)**

**QI=(observation – climatological average) / climatological standard deviation**

[Figure]

Figure R1: Quality Index (QI) of Sentinel3-OLCI chla images on (left) 4 and (right) 7 November 2021

**Lagrangian analysis of surface transports from Alcudia Bay**

[Figure]

Figure R2: Surface trajectories of model particles seeded in Alcudia Bay on 4 (top-left), 5 (top-right), 6 (bottom-left) and 7 (bottom-right) November. The initial and final positions are represented by green and red dots, respectively.

---

## Author Response (AR1)

**Point-by-point response to the reviewers´s comments.**

The reviewers´s comments are kept in black. Our responses are highlighted in red.

We thank both reviewers for their time and useful suggestions, which we think have contributed to improve the manuscript.

Review#1

This paper deals with an extreme upwelling event in the Balearic Islands (Mallorca) in November 2021 during the strong storm Blas. The authors first briefly analyse the results of a numerical model to describe the effects of this storm on coastal waters. They introduce different proxies (upwelling indices) to revisit 9 years of operational modelling results and characterise the extreme upwellings of the northern coast of Mallorca. A discussion is outlined on the specificity of these indices regarding the thermal stratification.

Despite a relatively short, simple and descriptive contribution, this study is scientifically sound and could be useful to scientists, modelers and policy makers in their interpretations of operational modelling. The paper is well written and clearly organised.

This is a good example of the benefits of operational oceanography. The time series (9 years) is probably to short and the tools not original but it opens the way to the analyze of this kind extreme event in the framework of climate change. I think this paper is suitable for publication as part of Copernicus Ocean state Report.

English is not my first language, so I have no particular skills in assessing English language proficiency.

We agree that the study is a good example of the benefits of operational oceanography and high-resolution forecast models. This idea is mentioned in the conclusion (l. 248-249 in the track change version).

We have also carefully checked the English language.

Finally, we do also recognize the importance of analyzing these events in the longer term when multi-decadal kilometer-scale reanalysis simulations become available in the study area. This sentence has been added at the end of the conclusion.

Review#2

The paper explores the signature of storms passages on localised upwelling on the Balearic Islands region. The authors make use of the Copernicus operational models and downstream regional high resolution model to characterise a recent event (storm Blas, November 2021) and compare it with modelled historical events back 9 years through the application of several upwelling indices definitions.

The manuscript is a good example of the use of Copernicus operational systems (models and observations) to identify and characterise events that would otherwise be difficult to capture by other oceanographic tools.

The methods used are broadly suitable for the focus of the project (e.g. model types and resolution) and the figures are clear and sufficiently informative. The organisation of the paper is also clear and well structured.

There are some aspects of the use of english that can be improved (some suggestions below, but maybe the authors can ask a native english speaker to revise it).

The authors compare the events found in the 9 year records of model outputs by the value of the four upwelling indices they calculated. While the focus of the manuscript seems to be on storm induced upwelling events, it would be useful if the authors offered an operational definition of an upwelling event for the historic record analysis. While upwelling intensity is a relevant metric, upwelling event duration is also a critical parameter. I would like to suggest the authors include a measure of it in their comparison. Furthermore, the four indices they use don't always agree on the presence of upwelling and maybe a brief critique of their sensitivity is needed. Finally, their interpretation of the effect of the storms on upwelling and subsequent enhancement of primary productivity is very speculative. First, chla is not a measurement of productivity (amount of assimilated carbon). plankton growth can be limited by nutrients, light or predation. Upwelling can aliviate nutrient limitation but I am guessing the storm passage would still induced light limitation. Additionally, chla retrieval near the coast and in the presence of resuspended sediments is bound to have large errors. It would be useful to see what associated errors are provided by the chla satellite scene (I believe the data now comes with pixel by pixed errors estimates?). Besides, the coastal embayment area of the NE of Mallorca island shows enhanced chla on 4november and the current fields could potentially have advected that water along the coast to contribute to the apparent chla enhancement on the 7th November (figure 2).

Maybe a short lagrangian particle simulation could be done to reject this hypothesis?

We agree that both intensity and duration are important parameters for the operational definition of upwelling events. As illustrated in Figure 4, the intensity of an upwelling event differs depending on the index used for its characterization, which constitutes a major difficulty for their identification. The intensity might be measured by the quantification of surface temperature gradients, offshore Ekman transport, offshore total transport, or alongshore currents, as described by the four indices considered in this study. In this short manuscript, we do not aim at providing a general operational definition of these upwelling events, which we think would require a dedicated analysis to carefully define specific thresholds for the different indices. Instead, we rather want to illustrate the sensitivity of this identification to the selected indices, focusing on the comparison of the most extreme storm-induced cases detected over the last 9 years. As suggested, we discuss this sensitivity in the conclusion of the revised manuscript (l. 234-237). We have also added duration metrics of the three main extreme events (Jan 2017, Jan 2020 and Nov 2021), computed from the total transport index (l. 237-241). This allows to provide elements for the future potential operational definition of such upwelling events.

Concerning the interpretation of ocean color images, we agree with the reviewer that they are certainly not exempt of uncertainties given the very coastal and cloudy situation which is considered. Unfortunately, there are no error estimates associated with these Sentinel 3 – OLCI CHLA observations so far. The only uncertainty metric that is provided with the data product is the so-called quality index (QI), which measures, pixel by pixel, the difference between the observation and the climatological average, normalized by the climatological standard deviation. The QI maps for 4 and 7 November 2021 are included in a new supplementary material (Figure R1).

[Figure]

*QI= (observation – climatological average) / climatological standard deviation*

On 7 November, in the area of enhanced Chla on the northwestern coast of Mallorca Island, the QI values range from 1 to 2, with a peak from 4 to 5 in the more coastal zone. Given the extreme conditions considered here and the very likely limitation of the climatological estimates in these coastal areas (the 300m-resolution Sentinel3-OLCI data are only available since 2016), these values seem to be realistic and, in our opinion, do not allow to identify evident errors in these observations.

We agree with the reviewer that CHLA observations in shallow coastal environments can be significantly affected by sediment resuspension. However, in our opinion, this is unlikely to be the case here due to the steep local topography that deepens to more than 500m a few km from the coast.

We have included these elements in the revised manuscript (l. 174-178), also tempering our interpretation of these ocean color images and highlighting the limitations due to the lack of any other complementary observations in the area during the study period.

In addition, following the reviewer´s suggestion, we have performed a short Lagrangian analysis to evaluate the hypothesis of a lateral transport of Chla from the northern Bay in Mallorca Island (Alcudia Bay) towards the upwelling area. The model didn't allow any surface transport exporting particles outside the Alcudia Bay on the days before and during the upwelling event. These new results allow us to reasonably reject this hypothesis. We have added this additional information in a supplementary material associated with the revised manuscript (Figure R2).

With these elements at hand, and within the limitations of both the model and observations, we believe that the spatial consistency between the observed patch of enhanced CHLA and the model upwelling area is probably indicative of an upwelling-induced local enhancement of CHLA.

Specefici suggestions:

L27: transports, "behind storm Gloria upwelling event" in ...

Done

L35: temperature "and" the second one "in terms of " cross-shore ...

Corrected

L54, 56, 68-69. When refering to upwelling as a process/phenomenon, use the singular , e.g. L56, wind induced upwelling is known to occur...

Corrected

L73-74: This study "was motivated by the" intense sea sureface [...] of the Balearic Islands "seen" in high-resolition...
Done

L77: ...singularity "within" the time-series ...

Done

L97: provide references for DA and atmospheric forcing similar to what you offer for the WMOP.

References have been added.

L109: As the vertical transport and nutrient fluxes are not used here I would remove its reference.

We have removed this part, but still kept this reference since it is relevant for the previous ítem. Also, we have reordered the sentence to better match with the following presentation of upwelling índices.

L142: the total "cross-shelf" model transport

Done

L169: Figure 2 "hints at " the enhancement of the ...
Corrected

Figure 4 caption: are daily upwelling indices daily averaged values?

The time series show 3-hourly values with a 24-hour running average. We have updated the legend accordingly to be more precise.

L223: To evaluate the complementarity of the upwelling indices you need an operational definition of upwelling event for each of the indices. I see many upwelling events as suggested by UIsst that don't appear in UIalongshore etc..

We have removed the term complementarity since we agree that it would require a more careful analysis. We rather highlight the differences between these indices, in particular for the three main storm-related events considered here (i.e. Blas, Gloria, Storm in January 2017). In particular, the magnitude of SST gradients or alongshore current velocity does not necessarily reveal the whole intensity of the underlying offshore transport.

L233: I don't think that Figure 2 provides confidence in the realism of the numerical results. Comparison with HF radar would.

The WMOP operational model predictions are assessed on a daily basis in the areas with available HF radar observations (Ibiza Channel, Ebro Delta and Gibraltar Strait; https://www.socib.es/?seccion=modelling&facility=wmedvalidation, Mourre et al., 2018). In particular, these predictions were proven to be realistic in the Ebro Delta region during storm Gloria (Sotillo et al., 2021). Unfortunately here, no HF radar data are available off the northwestern coast of Mallorca Island to perform such analysis. Notice also that WMOP routinely assimilates surface current observations from the HF radar installed in the Ibiza Channel, which might have a remote positive impact in the study area.

Again, within the limitations of both the model and observations, we believe that the spatial consistency between the CHLA enhancement area and the model upwelling area suggests that the model provides a reasonable representation of its spatial structure. We have clarified our statement in the revised manuscript.

L234: As I said above, I am not convinced given the evidence provided in the mansucript, that there is a real upwelling-induced enhancement of chla in the region.

See our explanation above.

Finally, we have also carefully checked the English language.

---

## Author Response (AR2)

**Point-by-point response to the notification to the authors from review file validation**

**Notification to the authors:**
**1. Checking your paper, I noticed that some of your tables contain coloured cells. Please note that this will not be possible in the final revised version of the paper due to HTML conversion of the paper. When revising the final version, you can use footnotes or italic/bold font. For now, the process will continue, but please note that the final version cannot be published by using coloured tables.**

**We have removed the color in Table1.**

**2. Your reference list includes works "submitted to". Such works can be cited upon submission if being available to the reviewers. They should not be cited in the final, accepted manuscript, unless published, accepted for publication, or available as preprint with a DOI.**

**We have updated the reference with the doi of the associated preprint.**

**Notice that this article is presently in the final phase of the review (with minor corrections) in the same issue (Ocean State Report) of the State of the Planet journal.**